# Evaluating cell lines as models for metastatic breast cancer through integrative analysis of genomic data

Ke Liu [1,2], Patrick A. Newbury [1,2], Benjamin S. Glicksberg[3], William Z.D. Zeng[3], Shreya Paithankar[4], Eran R. Andrechek[5] & Bin Chen [1,2]

Cell lines are widely-used models to study metastatic cancer although the extent to which they recapitulate the disease in patients remains unknown. The recent accumulation of genomic data provides an unprecedented opportunity to evaluate the utility of them for metastatic cancer research. Here, we reveal substantial genomic differences between breast cancer cell lines and metastatic breast cancer patient samples. We also identify cell lines that more closely resemble the different subtypes of metastatic breast cancer seen in the clinic and show that surprisingly, MDA-MB-231 cells bear little genomic similarities to basal-like metastatic breast cancer patient samples. Further comparison suggests that organoids more closely resemble the transcriptome of metastatic breast cancer samples compared to cell lines. Our work provides a guide for cell line selection in the context of breast cancer metastasis and highlights the potential of organoids in these studies.

[1] Department of Pediatrics and Human Development, College of Human Medicine, Michigan State University, Grand Rapids 49503 MI, USA. [2] Department of Pharmacology and Toxicology, College of Human Medicine, Michigan State University, Grand Rapids 49503 MI, USA. [3] Bakar Computational Health Sciences Institute, University of California San Francisco, San Francisco 94158 CA, USA. [4] Health Informatics and Bioinformatics, School of Computing and Information Systems, Grand Valley State University, Grand Rapids 49504 MI, USA. [5] Department of Physiology, Michigan State University, East Lansing 48824 MI, USA. Correspondence and requests for materials should be addressed to B.C. (email: Bin.Chen@hc.msu.edu)

Cancer cell lines were initially derived from tumors and cultured in a two-dimensional environment. Due to the merit of cell culture, they have been widely used as models to study cancer biology and test drug candidates[1]. However, the fact that many drugs with promising preclinical evidence fail in the clinic urges the reinvestigation of cell lines as tumor models[2]. The differences between cell lines and tumors have raised the critical question to what extent cell lines recapitulate the biology of tumor samples[3,4].

The emergence of large-scale genomic data provides an unprecedented opportunity to quantify the biological differences between cancer cell lines and human tumors. The Cancer Genome Atlas (TCGA) project characterized both genomic and transcriptomic profiles for more than 10,000 human tumor samples across over 32 tumor types[5]. The Cancer Cell Line Encyclopedia (CCLE) characterized both genomic and transcriptomic profiles for more than 1000 cell lines[6]. Domcke et al.[7] performed a comprehensive comparison of molecular profiles between 47 ovarian cancer cell lines and ovarian tumor samples and showed that several rarely used cell lines more closely resembled high-grade serous ovarian tumor samples than popular cell lines. We examined the transcriptome similarity between hepatocellular carcinoma (HCC) cell lines and HCC tumor samples and demonstrated that nearly half of the HCC cell lines did not resemble HCC tumor samples[8]. Jiang et al.[9] conducted a comprehensive comparison of molecular portraits between breast cancer cell lines and primary breast cancer samples, and uncovered both similar and dissimilar molecular features.

Cancer metastasis is the most common cause of cancer-related death, thus there is an urgent need of new drugs for treating cancer metastasis[10,11]. Previous cell line evaluation analysis was mainly performed in reference to primary tumors. It remains unknown whether cell lines closely resemble metastatic cancer and thus are appropriately used in translational research. Robinson et al.[12] performed whole-exome and transcriptome sequencing on 500 adult patients with metastatic solid tumors and recently released their dataset (MET500). This large-scale genomic profiling combined with existing genomic data allows the evaluation of the utility of cell lines as models for metastatic cancer. Using breast cancer as a case study, we comprehensively compare multiple types of molecular features between breast cancer cell lines and metastatic breast cancer samples (Fig. 1a–e). Based on our analyses, we identify cell lines that closely resemble the transcriptome of different subtypes of metastatic breast cancer samples. In addition, we evaluate patient-derived organoids and show their potential in preclinical studies. Our work provides useful guidance for choosing cell lines in metastasis-related translational research and could be easily extended to other cancer types.

## Results

**Comparison of genomic profiles.** We first compared somatic mutation profiles between MET500 breast cancer samples and CCLE breast cancer cell lines. Whole-exome sequencing was performed for MET500 breast cancer samples, while hybrid capture sequencing was performed for CCLE cell lines. We thus only focused on the 1630 genes genotyped in both studies. We were particularly interested in two types of genes that may play important roles in breast cancer metastasis: genes that are highly mutated in metastatic breast cancer and genes that are differentially mutated between metastatic and primary breast cancers.

Consistent with previous research, we identified a long-tailed mutation spectrum of the 1630 genes in MET500 breast cancer samples and 69 of them were highly mutated (mutation frequency >0.05; Supplementary Fig. 1a). The five most-altered genes were

*TP53* (0.67), *PIK3CA* (0.35), *TTN* (0.29), *OBSCN* (0.19), and *ESR1* (0.14). We identified 19 differentially mutated genes between MET500 and TCGA samples (FDR < 0.001) and the five most significant genes were *ESR1, TNK2, OBSCN, CAMKK2*, and *CLK1* (Supplementary Fig. 1b). Interestingly, all of these 19 differentially mutated genes had higher mutation frequency in MET500 than TCGA, which is consistent with a previous study showing that metastatic cancer has increased mutation burden compared to primary cancer[12]. Sixty-eight percent of them were also among the 69 highly mutated genes mentioned above. After merging the two gene lists, 75 unique genes remained (Fig. 2a and Supplementary Data 1). The median mutation frequency of the 75 genes across CCLE breast cancer cell lines was 0.07 and only 9% of them (*PRKDC, MAP3K1, TTN, ADGRG4, TP53, FN1*, and *AKAP9*) were mutated in at least 50% of cell lines, suggesting that majority of these gene mutations could be recapitulated by only a few cell lines. Surprisingly, 9 of the 75 genes (*ESR1, GNAS, PIKFYVE, FFAR2, RNF213, MYBL2, KAT6A, MAP4K4*, and *FMO4*) were not mutated in any cell line. Notably, *ESR1* has been identified as a driver gene of cancer metastasis and associated mutations can cause endocrine resistance of metastatic breast cancer cells[13,14], yet none of the cell lines could be used to appropriately model it.

We next asked whether there were genes specifically hypermutated in breast cancer cell lines. To address this question, we examined the mutation spectrum of the 32 genes that were mutated in at least 50% of the breast cancer cell lines. Surprisingly, 25 of them had low mutation frequency (<0.05) in MET500 breast cancer samples. Further analysis of somatic mutation profiles of the 25 genes in TCGA breast cancer samples confirmed their hypermutations were specific to breast cancer cell lines (Supplementary Fig. 1c).

In addition to somatic mutation spectrum, we also compared copy number variation (CNV) profiles between MET500 breast cancer samples and CCLE breast cancer cell lines. We observed a high correlation of median CNV values across the 1630 commonly genotyped genes (spearman rank correlation = 0.81; Fig. 2b). However, we also noticed that the gain-of-copy-number events in cell lines appeared to resemble metastatic breast cancer while loss-of-copy-number events did not. For the 711 genes showing copy-number-loss in CCLE breast cancer cell lines (median CNV < 0), their cell line derived median CNV values were significantly lower than that from MET500 breast cancer samples; however, no statistically significant difference was detected in the 919 genes with copy-number-gain (Supplementary Fig. 1d).

Out of the 57 CCLE breast cancer cell lines, 24 were derived from metastatic sites (Supplementary Data 2). We further divided the cell lines into two groups (according to whether derived from metastatic sites or not) and then compared the CNV profile of each group with MET500 breast cancer samples. We found cell lines derived from metastatic sites more closely resembled the CNV status of the 109 genes with high copy-number-gain (median CNV ≥ 0.4) in MET500 breast cancer samples (Fig. 2c, d and Supplementary Fig. 1e).

**Correlating cell lines with MET500 patient samples.** Transcriptome correlation analysis (TC analysis) is proven to be an effective approach to evaluate cell lines for research purpose[7,8,15]. Therefore, we performed TC analysis and ranked all 1019 CCLE cell lines according to their transcriptome similarity with MET500 breast cancer samples (see Methods). The top 20 cell lines were all breast cancer cell lines, suggesting metastatic breast cancer cells retain the transcriptomic signature from the tissue they originated in and cell lines have the potential to resemble the

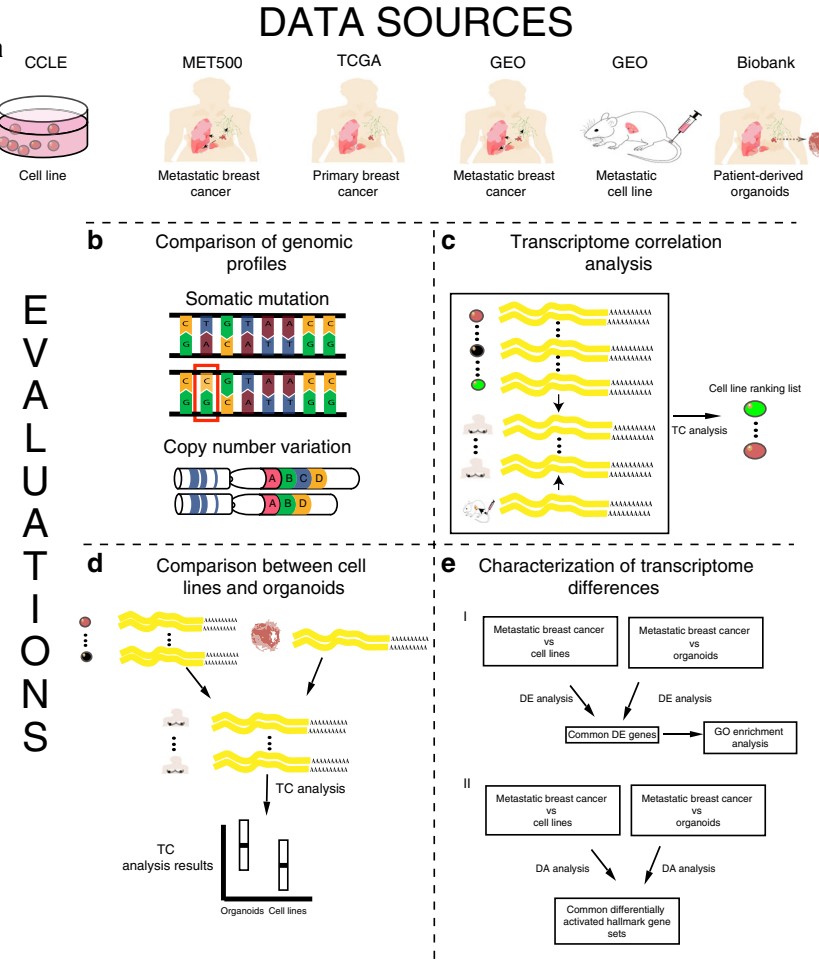

**Fig. 1** Overall research design. **a** Data sources and sample types used in this study. **b–e** Evaluations performed in this study: **b** comparison of genomic profiles, **c** transcriptome correlation analysis, **d** comparison between cell lines and organoids, and **e** characterization of transcriptome differences. TC analysis: transcriptome correlation analysis; DE analysis: differential gene expression analysis; DA analysis: gene set differential activity analysis

transcriptome of them (Fig. 3a). MDA-MB-415 and HMC18 were the two breast cancer cell lines that had highest and lowest transcriptome similarity, respectively (Spearman rank correlation of 0.415 and of 0.087).

We next assessed whether cell lines resembling the transcriptome of samples from different metastatic sites were identical. We were only able to consider liver and lymph node (the two sites which have at least nine samples) due to the lack of enough samples from other sites in the MET500 dataset. For each of them, we performed metastatic-site-specific TC analysis (i.e., compute transcriptome similarity of cell lines with samples derived from a specific metastatic site) and found the results were highly correlated (Fig. 3b) with MDA-MB-415 being the most-correlated cell line for both sites. In addition, we detected no statistically significant difference in expression correlation (with MDA-MB-415) between the two sites (Supplementary Fig. 2a).

Given the genomic heterogeneity of breast cancer, we further asked whether cell lines resembling the transcriptome of metastatic breast cancer of different subtypes were identical. To address this question, we first determined the PAM50 subtype of MET500 breast cancer samples with R package genefu and then applied t-SNE to visualize them (Fig. 3c). We found Basal-like samples clustered together and separated from other subtypes; additionally, the majority of LuminalA/LuminalB/Her2-enriched/ Normal-like samples were mixed together except two skin-derived samples. HER2-enriched samples seemed to be separated

from LuminalA/LuminalB samples but the boundary was not clear. These results suggested that subtype information was well maintained in metastatic breast cancer samples and additionally confirmed the feasibility of using PAM50 for subtyping metastatic breast cancer though it was initially developed using primary breast cancer data. We further confirmed the subtyping results by performing the same analysis on a combined dataset which contains both MET500 and TCGA breast cancer samples (Supplementary Fig. 2b). Next, we performed subtype-specific TC analysis (i.e., compute transcriptome similarity of cell lines with samples of a specific subtype) and found high correlation within LuminalA/LuminalB/Her2-enriched subtypes, in contrast to their relatively lower correlation to Basal-like subtype (Fig. 3d).

To confirm the robustness of our TC analysis results derived from the comparison between CCLE and MET500 RNA-Seq data, we downloaded CCLE gene expression data profiled by microarray and then searched the GEO database and assembled a microarray dataset containing the expression values of another 103 independent metastatic breast cancer samples. We repeated the TC analysis with microarray data and found results obtained from the two platforms were highly consistent with each other. First, there was a large overlap of the top-ranked cell lines. Out of the 10 cell lines having highest transcriptome similarity with the 103 metastatic breast cancer samples, 6 of them were within the 10 cell lines having highest transcriptome similarity with MET500 breast cancer samples. Second, both metastatic-site-specific and

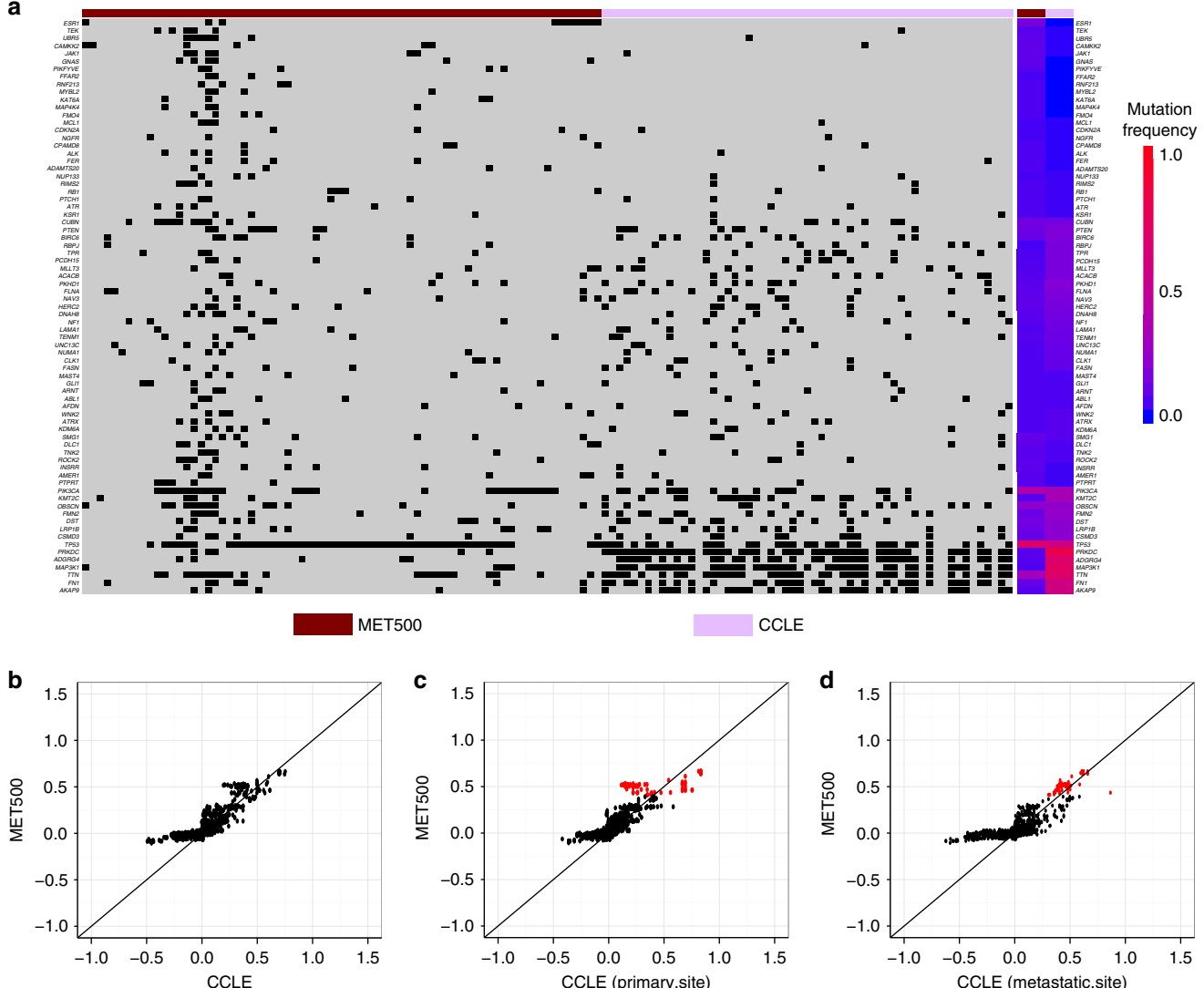

**Fig. 2** Comparison of genomic profiles between MET500 breast cancer samples and CCLE breast cancer cell lines. **a** Somatic mutation profile of the 75 genes across MET500 breast cancer samples and CCLE breast cancer cell lines. The top-side color bar indicates data sources (MET500 or CCLE) and the right-side color bar indicates mutation frequency. **b** Comparison of CNV profiles between MET500 breast cancer samples and CCLE breast cancer cell lines. **c** Comparison of CNV profiles between MET500 breast cancer samples and the 33 primary-site derived CCLE breast cancer cell lines. **d** Comparison of CNV profiles between MET500 breast cancer samples and the 24 metastatic-site-derived CCLE breast cancer cell lines. In panels **b**–**d**, each dot is a gene, with y-axis representing its median CNV value across MET500 breast cancer samples, and x-axis representing its median CNV value across CCLE breast cancer cell lines. In panels **c** and **d**, genes with high copy-number-gain in MET500 breast cancer samples were marked by red. Source data are provided as a Source Data file

subtype-specific TC analysis results showed high correlations (Supplementary Fig. 3). Due to such high consistency, it is not surprising that we observed similar correlation trends in metastatic-site-specific (and subtype-specific) TC analysis results (Supplementary Figs. 4 and 5).

About 24% of the 103 samples in the microarray dataset was derived from bone. Remarkably, the metastatic-site-specific TC analysis result of bone showed lower correlation with other sites (Supplementary Fig. 4). To exclude the possibility that this was caused by tumor purity issues, we applied ESTIMATE[16] on the microarray dataset and found the tumor purity of bone-derived samples was not significantly lower than that of liver, lymph node, and lung (Supplementary Fig. 6). Our results may not be too surprising given the fact that bone provides a very unique microenvironment including enriched expression of osteolytic genes[17]; however, this result needs to be confirmed in the future as more data become available.

**Subtype-specific cell line evaluation**. We attempted to identify cell lines which closely resemble the transcriptome of different subtypes of metastatic breast cancer based on the results of subtype-specific TC analysis. Given a subtype, we noticed that for a random CCLE cell line, its transcriptome similarity with MET500 breast cancer samples of that subtype approximately followed a normal distribution (Supplementary Fig. 7). Therefore, those breast cancer cell lines showing significantly higher transcriptome similarity were of our interest. Driven by this finding, for each subtype we first fit a normal distribution (the null distribution) with the transcriptome similarity values (derived from subtype-specific TC analysis) of all non-breast-cancer cell lines and then assigned each of the 57 breast cancer cell lines a right-tailed p-value. The most significant cell lines for LuminalA, LuminalB, Her2-enriched, and Basal-like subtypes were MDA-MB-415 (p-value = 3.59e-05), BT483 (p-value = 2.22e-07), EFM192A (p-value = 0.11e-03), and HCC70 (p-value = 0.40e-03), respectively. Using criteria of FDR ≤ 0.01, we identified 20, 28, and

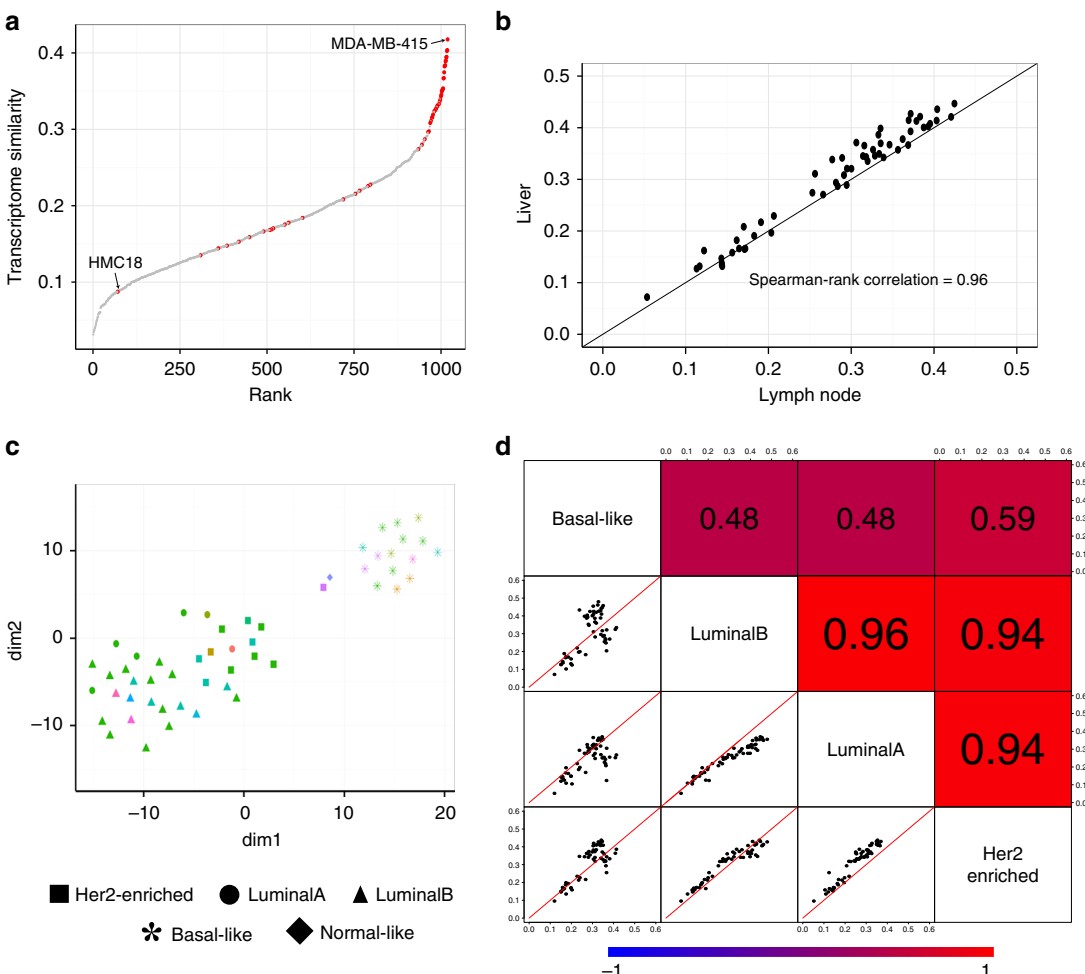

**Fig. 3** TC analysis between MET500 breast cancer samples and CCLE breast cancer cell lines. **a** In total, 1019 CCLE cell lines are ranked according to their transcriptome similarity with MET500 breast cancer samples. Each dot is a CCLE cell line and breast cancer cell lines are marked by red. **b** Metastatic-site-specific TC analysis results are highly correlated between liver and lymph node. Each dot is a CCLE breast cancer cell line, with x-axis representing its transcriptome similarity with the nine lymph-node-derived MET500 breast cancer samples, and y-axis representing its transcriptome similarity with the 27 liver-derived MET500 breast cancer samples. **c** t-SNE plot of MET500 breast cancer samples. Metastatic-sites are labeled by color and subtypes are labeled by shape. **d** Pair-wise comparison of subtype-specific TC analysis results. In the lower-left plots, each dot is a CCLE breast cancer cell line, with the two axis representing transcriptome similarity of the cell line with MET500 breast cancer samples of the two intersecting subtypes. The upper-right shaded values are the corresponding pair-wise spearman rank correlation values of each pair. Source data are provided as a Source Data file

24 significant cell lines for LuminalA, LuminalB, and Her2-enriched subtypes, respectively. Notably, most of these significant cell lines were derived from metastatic sites and 18 were shared by the three subtypes. Surprisingly, no cell line passed the criterion for Basal-like subtype. We further examined whether this was due to the limited number of Basal-like MET500 breast cancer samples, but found that the number of LuminalA samples was even less than that of Basal-like samples. After we used a more loosened FDR cutoff of 0.05, we obtained 22 significant cell lines for Basal-like subtype. All statistical testing results are listed in Supplementary Data 3.

We next searched PubMed to examine the popularity of the 57 breast cancer cell lines (see Methods and Supplementary Data 2). MCF7 is most commonly used in metastatic breast cancer research (43.6% of total PubMed citations). In our analysis, although it was a significant cell line (according to criteria FDR ≤ 0.01) for LuminalB subtype, it was less correlated with LuminalB MET500 breast cancer samples than BT483 (Supplementary Fig. 8a). Following MCF7 is MDA-MB-231 (40.2% of total PubMed citations); however, it was not a significant cell line for any subtype. The third most commonly used cell line was T47D (3.9% of total PubMed citations), which was a significant cell line

for LuminalA and Her2-enriched subtypes. It did not show significantly lower correlation with LuminalA MET500 breast cancer samples than MDA-MB-415 (Supplementary Fig. 8b); however, compared to EFM192A, it was significantly less correlated with Her2-enriched MET500 breast cancer samples (Supplementary Fig. 8c).

We further explored cell line MDA-MB-231, one of the most widely used triple-negative cell lines in metastatic breast cancer research. We ranked all of the 1019 CCLE cell lines according to their transcriptome similarity with Basal-like MET500 breast cancer samples and the rank of MDA-MB-231 was 583. Consistent with this, MDA-MB-231 was significantly less correlated with Basal-like MET500 breast cancer samples than HCC70 (Fig. 4a). We observed similar patterns with CNV data (Fig. 4b). We also examined how MDA-MB-231 recapitulated the somatic mutation spectrum of Basal-like metastatic breast cancer samples and found only three of the 25 highly mutated genes (mutation frequency ≥ 0.1 in Basal-like MET500 breast cancer samples) were mutated in MDA-MB-231 (Fig. 4c). Since CCLE data for MDA-MB-231 was generated in vitro, to confirm our finding we obtained another independent microarray dataset

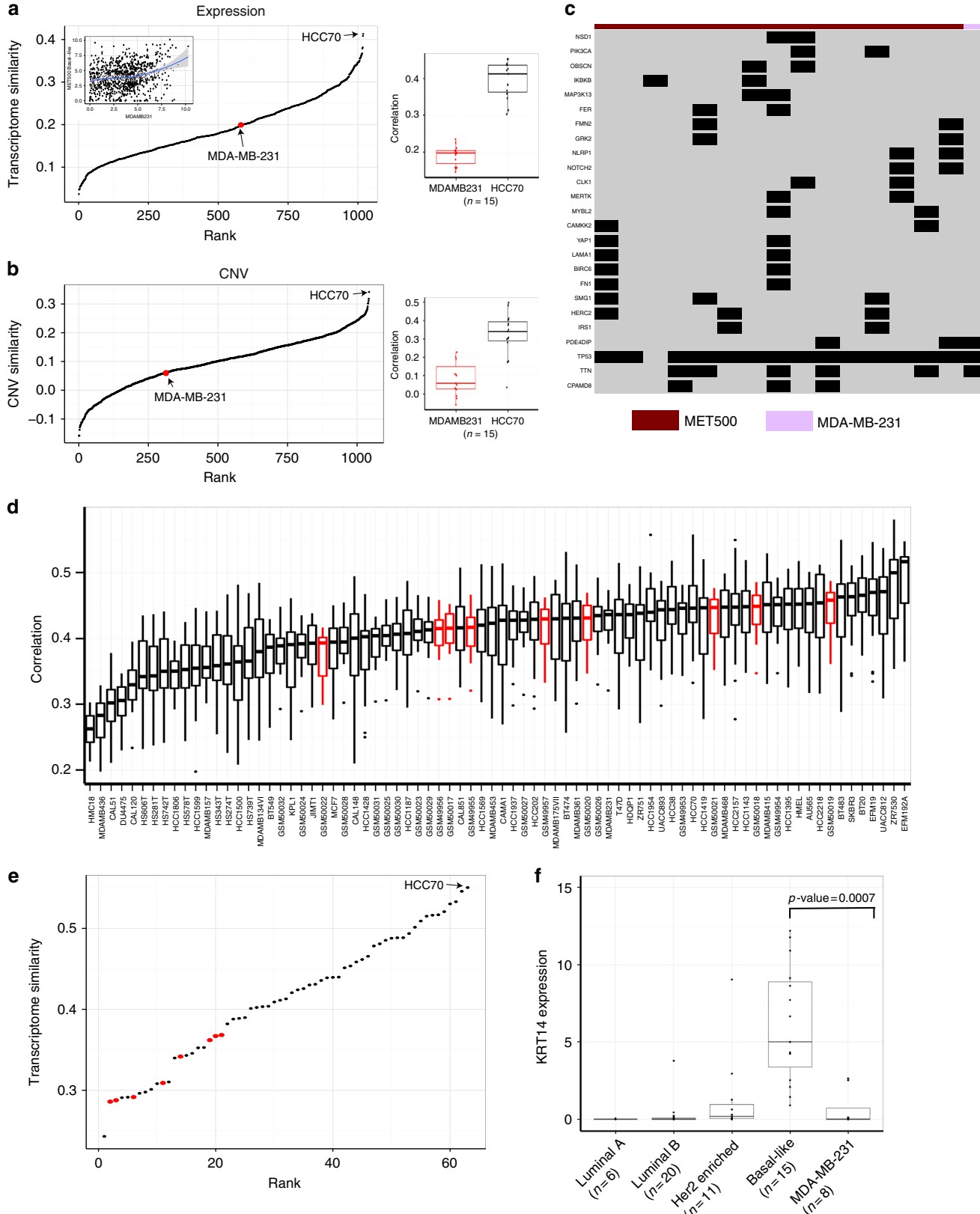

which profiled the gene expression of MDA-MB-231 cell lines derived from lung metastasis in vivo[18]. We found, however, that even these MDA-MB-231 cell lines in vivo did not most closely resemble the transcriptome of lung metastasis breast cancer samples. The breast cancer cell line which showed highest correlation with lung metastasis breast cancer samples was EFM192A (Fig. 4d).

To further confirm the low transcriptome similarity of MDA-MB-231 with Basal-like MET500 breast cancer samples, we re-processed the RNA-Seq data of CCLE breast cancer cell lines

**Fig. 4** MDA-MB-231 has substantial genomic difference with Basal-like metastatic breast cancer samples. **a** The left panel shows the ranking of all 1019 CCLE cell lines according to their transcriptome similarity with Basal-like MET500 breast cancer samples. The top-left scatter plot within the first panel shows the expression of the 1000 genes used in TC analysis, with the x-axis representing expression value in MDA-MB-231, and the y-axis representing median expression value across Basal-like MET500 breast cancer samples. The boxplot on the right shows the distribution of expression correlation with Basal-like MET500 breast cancer samples for MDA-MB-231 and HCC70. In each box, the central line represents the median value and the bounds represent the 25th and 75th percentiles (interquartile range). The whiskers encompass 1.5 times the interquartile range. **b** The left panel shows the ranking of all 1019 CCLE cell lines according to their CNV similarity with Basal-like MET500 breast cancer samples; the boxplot on the right shows the distribution of CNV correlation with Basal-like MET500 breast cancer samples for MDA-MB-231 and HCC70. In each box, the central line represents the median value and the bounds represent the 25th and 75th percentiles (interquartile range). The whiskers encompass 1.5 times the interquartile range. **c** Somatic mutation profile of the 25 highly mutated genes across MDA-MB-231 and Basal-like MET500 breast cancer samples. **d** Boxplot of expression correlation between cell lines and lung-derived metastatic breast cancer samples. This includes CCLE breast cancer cell lines and lung-metastasis-derived MDA-MB-231 (colored red). In each box, the central line represents the median value and the bounds represent the 25th and 75th percentiles (interquartile range). The whiskers encompass 1.5 times the interquartile range. **e** Ranking CCLE breast cancer as well as additional seven MDA-MB-231 cell lines according to their transcriptome similarity with MET500 Basal-like breast cancer samples. Each dot is a cell line; the x-axis represents rank and the y-axis represents transcriptome similarity with Basal-like MET500 breast cancer samples. MDA-MB-231 cell lines were colored red. **f** Boxplot of KRT14 expression in MET500 breast cancer samples and MDA-MB-231 cell lines. The p-value is computed with the two-sided Wilcoxon rank-sum test. In each box, the central line represents the median value and the bounds represent the 25th and 75th percentiles (interquartile range). The whiskers encompass 1.5 times the interquartile range. Source data are provided as a Source Data file

(with the pipeline used to process MET500 RNA-Seq data); in addition, we also re-processed the RNA-Seq data of another seven MDA-MB-231 cell line samples collected from SRA database. We re-performed TC analysis between the breast cancer cell lines (with re-processed data) and Basal-like MET500 breast cancer samples and drew similar conclusion as our previous analysis (Fig. 4e). Recently, Nguyen et al.[19] performed single cell RNA-Seq on human breast epithelial cells and confirmed that KRT14 expression was a hallmark of Basal cells. Strikingly, we found KRT14 expression in the eight MDA-MB-231 cell line samples was significantly lower than that of Basal-like MET500 breast cancer samples (p-value = 0.0007); however, such significant KRT14 differential expression was not detected between MDA-MB-231 and MET500 breast cancer samples of non-Basal-like subtypes (Fig. 4f). Our analysis indicates that although MDA-MB-231 was classified as Basal-like subtype and bears the triple-negative phenotype, its cell type of origin may not be Basal cell, which could partially explain why its genomic profile is substantially different from Basal-like MET500 breast cancer samples.

**Comparing cell lines with organoids**. Owing to the advancement of three-dimensional (3D) culture technology, more and more tumor patient-derived organoids have been established and widely used in translational research[20,21]. However, their utility to model metastatic cancer has not been comprehensively evaluated with large-scale genomic data. To fill this gap, we performed additional TC analysis on 26 patient-derived breast cancer organoids using RNA-Seq data. The aforementioned subtype-specific TC analysis showed that the Basal-like subtype had relatively lower correlation with other subtypes and we also observed similar trend in organoids (Fig. 5a). We next asked whether organoids outperformed cell lines in resembling the transcriptome of metastatic breast cancer. For each of the non-Basal-like organoids, we computed its transcriptome similarity with non-Basal-like MET500 breast cancer samples and found organoids had significantly higher transcriptome similarity than CCLE breast cancer cell lines (Fig. 5b, left panel). The superiority of organoids was also observed in the TC analysis of Basal-like subtype (Fig. 5b, right panel). The previous analysis revealed that MDA-MB-415, BT483, and EFM192A were the three most significant cell lines for LuminalA, LuminalB, and Her2-enriched subtypes, respectively. Interestingly, for all the three subtypes MMC01031 was the organoid showing highest transcriptome similarity and had significantly higher correlation with MET500

breast cancer samples than the corresponding most significant cell line. Organoid W1009 had the highest transcriptome similarity with Basal-like MET500 breast cancer samples and the expression correlation values were also significantly higher than HCC70, the triple-negative cell line that is most significant for Basal-like subtype (Fig. 5c).

**Transcriptome differences between models and patients**. Our TC analysis has shown that in vitro models such as cell lines and organoids could resemble the transcriptome of metastatic breast cancer to some extent. However, they are still different in many aspects. To characterize such differences, we performed differential gene expression analysis among MET500 breast cancer samples, CCLE breast cancer cell lines, and organoids (Supplementary Fig. 9). For non-Basal-like subtypes, 2348 genes (2143 up-regulated, 205 down-regulated) were identified as differentially expressed in both MET500-vs-CCLE and MET500-vs-organoids comparisons. For Basal-like subtype, there were 1378 common differential expressed (DE) genes (1117 up-regulated, 261 down-regulated). After intersecting the above two common DE gene lists, we finally obtained 1017 subtype-and-model-independent DE genes (947 up-regulated, 70 down-regulated) and then performed GO enrichment analysis. For the 947 up-regulated ones, 29 GO terms were identified as significant (FDR < 0.001) and most of them were immune-related, illustrating the large gap between culture media and tumor microenvironment (Supplementary Data 4). The two terms "platelet degranulation" and "chemotaxis" were also detected as significant. Besides microenvironment, our results also implicated the difference of intrinsic characteristics between metastatic breast cancer cells and in vitro models. For example, the enrichment on "steroid metabolic process" suggested that both cell lines and organoids may not sufficiently resemble the reprogrammed metabolism of metastatic breast cancer. Surprisingly, for the 70 down-regulated subtype-and-model-independent DE genes, no GO terms passed the FDR < 0.001 criteria, which could be due to the small gene number. We decreased the FDR cutoff to 0.1 and observed four significant terms with "dna replication" being the most significant (FDR = 0.037). To further confirm that batch effects were not dominating DE analysis results, we used RUVg[22] to infer the values of hidden factors ($k = 1$) and re-performed DE analysis mentioned above and identified 749 subtype-and-model-independent DE genes, all of which were among the previously identified 1017 subtype-and-model-independent DE genes.

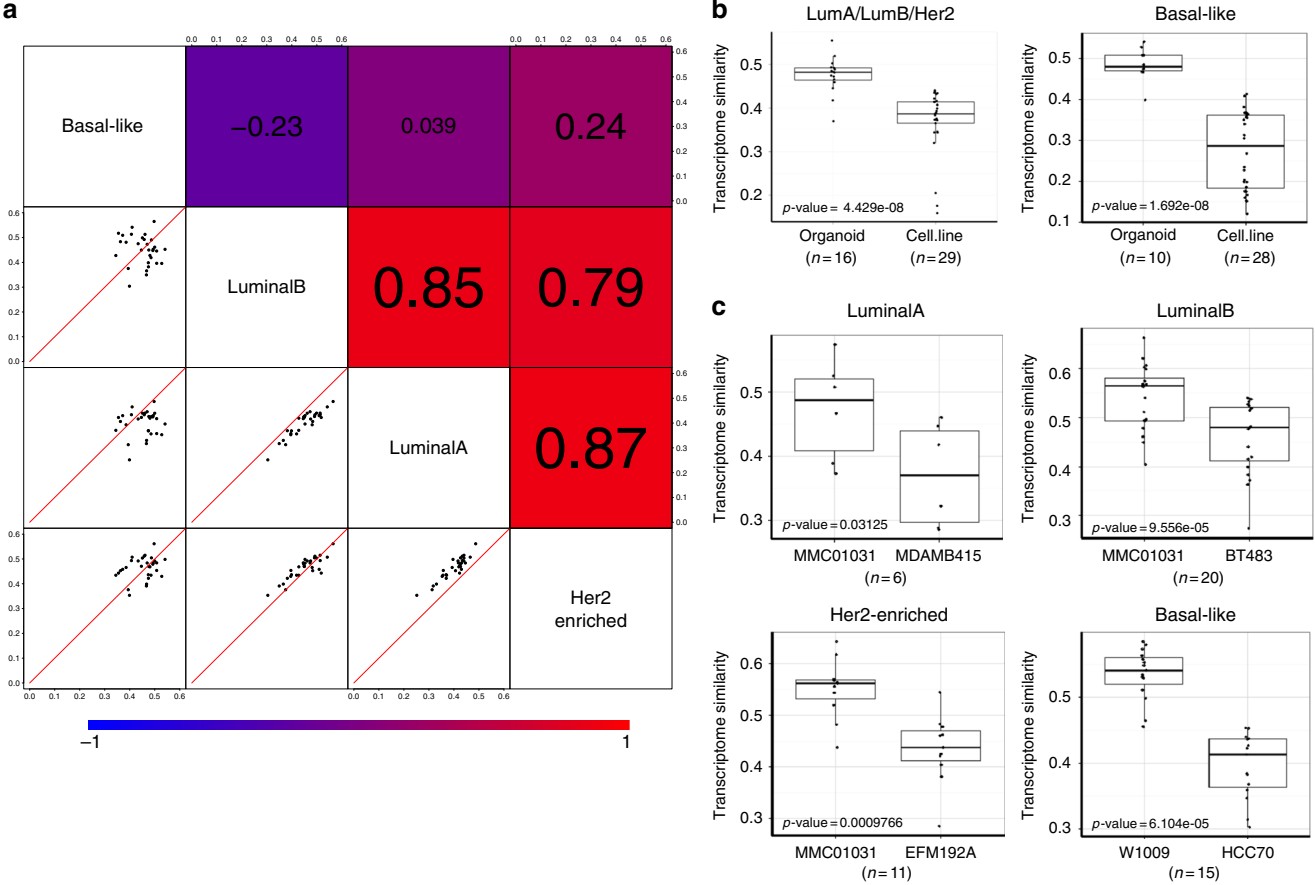

**Fig. 5** Comparing CCLE breast cancer cell lines with patient-derived organoids using gene expression data. **a** Pair-wise comparison of subtype-specific TC analysis results. In the lower-left plots, each dot is an established organoid, with the two axis representing transcriptome similarity of the organoid with MET500 breast cancer samples of the two intersecting subtypes. The upper-right shaded values are the corresponding pair-wise spearman rank correlation values of each pair. **b** Boxplot of transcriptome similarity (with MET500 breast cancer samples of different subtypes) of CCLE breast cancer cell lines and organoids. *P*-values are computed with the two-sided Wilcoxon rank-sum test. In each box, the central line represents the median value and the bounds represent the 25th and 75th percentiles (interquartile range). The whiskers encompass 1.5 times the interquartile range. **c** For each subtype, the most-correlated organoid has significantly higher expression correlation with MET500 breast cancer samples of that subtype than the most-correlated cell line. *P*-values are computed with the two-sided Wilcoxon rank-sum test. In each box, the central line represents the median value and the bounds represent the 25th and 75th percentiles (interquartile range). The whiskers encompass 1.5 times the interquartile range. Source data are provided as a Source Data file

We further performed gene set differential activity (DA) analysis on the 50 MSigDB hallmark gene sets to characterize differences regarding specific biological processes (Fig. 6a, Supplementary Fig. 10). For non-Basal-like subtype, we identified 35 and 32 significant gene sets in MET500-vs-CCLE and MET500-vs-organoids comparisons, respectively (FDR < 0.001; Supplementary Data 5). There were 26 significant gene sets in common and for 23 of them the *p*-values derived from MET500-vs-CCLE comparison were lower than that derived from MET500-vs-organoid comparison, which may be unsurprising given that organoids more closely resemble the transcriptome of metastatic breast cancer samples (Fig. 6b, left panel). We also performed DA analysis for Basal-like subtype, identifying 19 and 24 significant gene sets in MET500-vs-CCLE and MET500-vs-organoids comparisons, respectively (Fig. 6b, right panel). For each of the subtypes, we classified the 50 hallmark gene sets into four categories according to DA analysis results:

- Category 1: Only significant in MET500-vs-organoids comparison (e.g., ANDROGEN RESPONSE).
- Category 2: Only significant in MET500-vs-CCLE comparison (e.g., E2F TARGETS).
- Category 3: Significant in both MET500-vs-organoids and MET500-vs-CCLE comparisons (e.g., COMPLEMENT).
- Category 4: Not significant in either comparison (e.g., FATTY ACID METABOLISM).

Interestingly, 27 gene sets could be consensually classified into one specific category, regardless of the subtype. Figure 6c shows the distribution of ssGSEA scores of the representative gene set for each category.

## Discussion

In cancer research, cell lines have been traditionally used to test drug candidates and study disease mechanism. The genomic profile comparison showed that breast cancer cell lines poorly recaptured somatic mutation patterns of metastatic breast cancer samples, while their CNV profiles were more consistent. Moreover, it is worth noting that cell lines carried many specific genomic alternations, possibly due to culture effects. Examples included the 25 genes presenting cell-line-specific hypermutation. Such large genomic differences and variations revealed by the comparison indicates the importance of selecting cell lines to represent heterogeneous metastatic cancer samples. This study

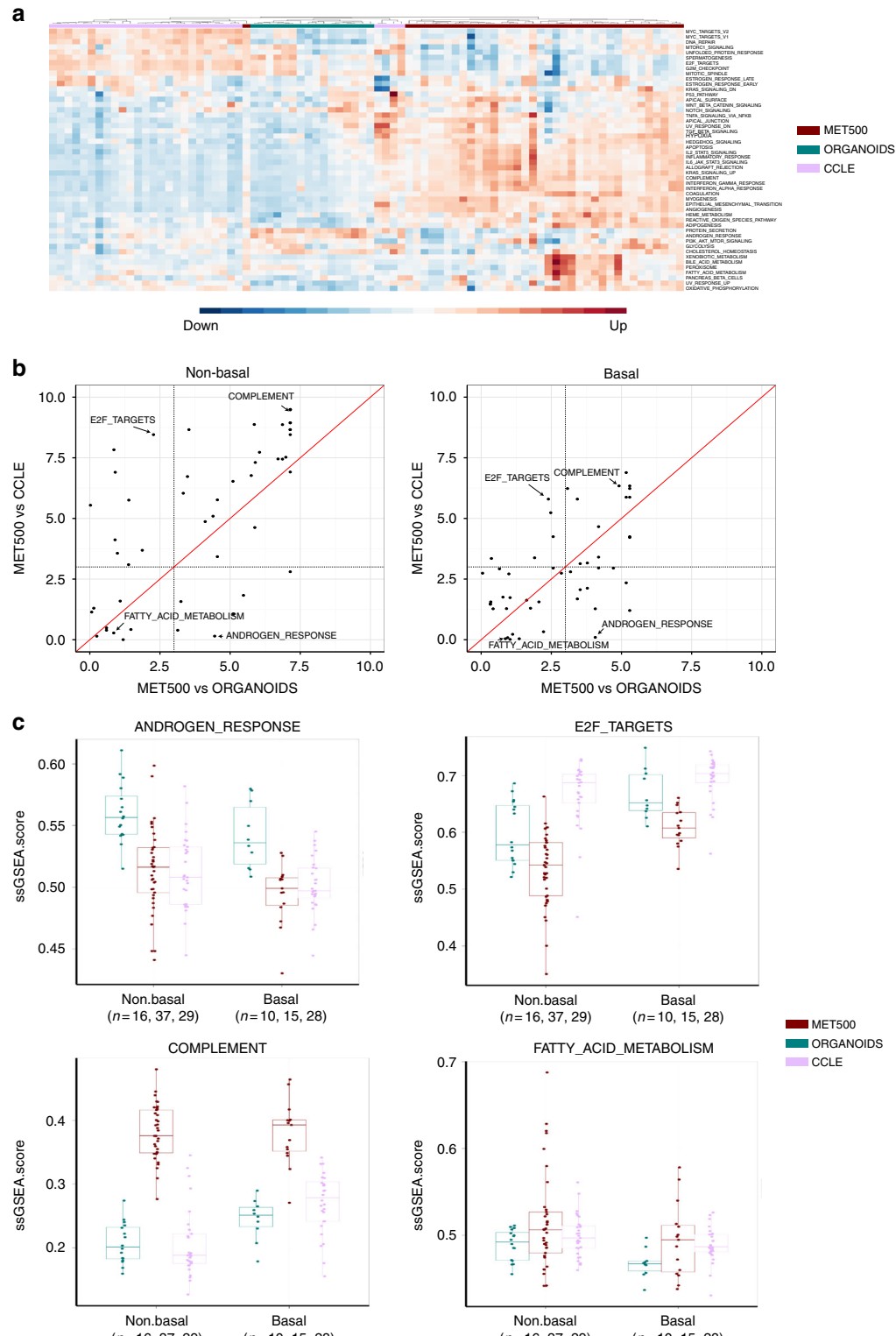

**Fig. 6** Comparison of ssGSEA scores of the 50 MSigDB hallmark gene sets. **a** Visualization of ssGSEA scores across non-Basal-like CCLE breast cancer cell lines, MET500 breast cancer samples, and organoids. **b** DA analysis results of different breast cancer subtypes. Each dot is a hallmark gene set, with the x-axis representing −log10(FDR) derived from MET500-vs-organoids comparison, and the y-axis representing −log10(FDR) derived from MET500-vs-CCLE comparison. **c** Boxplot of ssGSEA scores of the four representative gene sets. In each box, the central line represents the median value and the bounds represent the 25th and 75th percentiles (interquartile range). The whiskers encompass 1.5 times the interquartile range. Source data are provided as a Source Data file

investigated two important factors (i.e., metastatic site and cancer subtype) that need to be considered during cell line selection.

Metastatic sites have their distinct microenvironment that has a large impact in shaping the genomic profiles of metastatic cancer cells. However, the metastatic-site-specific TC analysis did not identify cell lines with metastatic-site-specific utility, which seems not to reflect the impact of microenvironment. Further differential gene expression analysis revealed higher expression of immune-related genes in metastatic cancer samples (compared to cell lines), suggesting that the media used to culture cancer cell lines did not model tumor microenvironment appropriately. Therefore, we conclude cell lines evaluated in this study do not carry indicative genomic signatures that are shaped by the microenvironment of individual metastatic sites and that might be the reason why we did not find metastatic-site-specific cell lines.

Breast cancer is quite heterogeneous and we showed that PAM50 subtypes were maintained in metastatic breast cancer cells. Considering the large genomic difference between Basal-like and other subtypes, it is not surprising that in subtype-specific TC analysis Basal-like subtype showed lower correlation with others. Our analysis reveals the importance and necessity of subtype-specific cell line selection. In the future as data continue to accumulate, more factors can be considered for appropriate cell line selection and we can start building an ad-hoc mapping algorithm: inputs would be the characteristics of metastatic cancer samples (subtype, metastatic site, even age, race, stage, etc.) as well as the specific scientific question of interest and the output would be a list of appropriate cell lines.

Surprisingly, we found MDA-MB-231, the widely used triple-negative cell line in metastatic breast cancer research, was dramatically different from Basal-like metastatic breast cancer samples. According to our analysis, HCC70 seems to be a better model, but this does not mean it can be directly employed to study cancer metastasis as many other criteria are needed for the assessment. Triple-negative breast cancer is itself highly heterogeneous. In a recent study, Nguyen et al.[19] showed that breast epithelial cell populations corresponded to breast cancer subtypes. Previous researchers have associated MDA-MB-231 with Claudin-low subtype and our analysis suggested that the cell type of origin of MDA-MB-231 cell line may not be Basal cells[23]. In the future, as more single-cell RNA-Seq data become available, it would be valuable to accurately determine the cell type of origin of MDA-MB-231 and optimize its usage in cancer metastasis-related studies.

Organoids are recently established using 3D culture methods. Our analysis suggested that compared to cell lines, they resemble the transcriptome of patient samples more closely, which is a critical characteristic in drug testing. It is also important to note that cell lines evaluated in our study were established much earlier than organoids. During the culturing process, they could have accumulated additional genomic alternations, which may partially explain why organoids are more tightly correlated with patient samples. Recent studies have shown that organoids preserve the histological architecture, gene expression, and genomic landscape of the original tumor[24]. Together with our comparative studies, we conclude that although the value of organoids in translational research has not been fully recognized, while their high genomic similarity with patient samples warrants further investigation.

Prior to this study, researchers have proposed different computational methods to measure the similarity between cell lines and patient samples[7,9]. It has been shown that gene expression is one of the most informative features to predict drug response and weighting cell lines based on their transcriptome similarity with patient samples increases predictive power in gene expression-based drug discovery[8,25–27]. Therefore, in this paper we ranked cell lines according to TC analysis results. Note that different expression features could be employed to measure transcriptome similarity depending on the question. For example, for researchers who specifically focused on the molecular mechanisms of cancer metastasis, it may not be necessary to require the cell line model (such as MDA-MB-231) to resemble the whole transcriptome of patient samples as long as it could mimic the key biological processes in cancer metastasis. In such scenario, ranking cell lines according to their invasiveness might be more appropriate.

In summary, by leveraging publicly available genomic data, we comprehensively evaluated the utility of breast cancer cell lines as models for metastatic breast cancer. Our study introduces a simple framework for cell line selection which can be easily extended to other cancer types. Although there are concerns about data quality and discrepancies between different studies/platforms, our large-scale analysis and cross-platform validation hopefully addresses these concerns and demonstrates the power of leveraging open data to gain biological insights of cancer metastasis. We hope that the recommendations in this study may facilitate improved precision in selecting relevant cell lines for modeling in metastatic breast cancer research, which may accelerate the translational research.

## Methods

**Datasets.** The raw RNA-Seq data of MET500 breast cancer samples (diagnosed as "Breast Invasive Ductal Carcinoma") were downloaded from dbGap (under accession number phs000673.v2.p1) and further processed using RSEM[28,29]. FPKM values were used as gene expression measure. To keep consistent with other RNA-Seq datasets, only the RNA-Seq samples profiled with PolyA protocol were considered. The somatic mutation and CNV data of MET500 samples were downloaded from MET500 web portal (https://met500.path.med.umich.edu/downloadMet500DataSets).

All CCLE pre-processed data (including gene expression profiled by RNA-Seq and microarray, somatic mutation and CNV) were downloaded from the CCLE data portal (https://portals.broadinstitute.org/ccle). The raw RNA-Seq data of 55 CCLE breast cancer cell lines were downloaded from GDC Legacy Archive (https://portal.gdc.cancer.gov/legacy-archive/search/f; HMEL and HS274T were missing due to unknown reasons) and further processed by RSEM. For each CCLE breast cancer cell line, we computed spearman rank correlation between gene expression values quantified by CCLE pipeline and RSEM. Supplementary Fig. 11a shows the distribution of the derived spearman rank correlation values. The median of the distribution is 0.9, suggesting that the gene expression values quantified by the two pipelines are highly consistent. As an example, Supplementary Fig. 11b shows such consistency in MCF7 cell line. We noticed that for MET500 cohorts both tumor and matched normal DNA were profiled in exome sequencing while for CCLE cell lines the somatic mutation was called by MuTect2 using a mode that does not require matched normal DNA[6,30]. Therefore, in our analysis we used the filtered version of CCLE somatic mutation MAF file (CCLE_hybrid_capture1650_hg19_NoCommonSNPs_NoNeutralVariants_CDS _2012.05.07.maf) in which common polymorphism variants have been excluded.

For TCGA Breast Invasive Ductal Carcinoma samples, somatic mutation calling results were downloaded from cBioPortal[31,32] on 12 April 2018 (using R package cgdsr) and RSEM-processed gene expression data were downloaded from UCSC Xena data portal (https://xena.ucsc.edu/)[33].

The RNA-Seq data of patient-derived organoids were from BC Organoids Biobank[34].

We searched GEO and manually assembled a microarray dataset containing gene expression values of 103 metastatic breast cancer samples[35–38]. The GEO accession numbers used were GSE11078, GSE14017, GSE14018, and GSE54323.

The gene expression data of lung-metastasis-derived MDA-MB-231 were downloaded from GEO under accession number GSE2603. The RNA-Seq data of non-CCLE MDA-MB-231 were downloaded from SRA and the accession numbers are ERR1982279, ERR1982280, ERR2022825, SRR2532366, SRR4822549, SRR6451704, and SRR6451705.

Detailed statistics of the above datasets are listed in Supplementary Data 6.

**Gene filtering.** We downloaded the list of 1650 genes covered by CCLE hybrid capture sequencing from CCLE data portal (https://data.broadinstitute.org/ccle_legacy_data/hybrid_capture_sequencing/CCLE_hybrid_capture1650_HGNC_info_2012.02.20.txt). Then, we applied the following steps to get the final 1630 highly confident genes.

 

1. Ten gene symbols which do not have associated HGNC records were removed.
2. Nine gene symbols which do not have associated RefSeq records (downloaded from UCSC genome browser, hg19) were removed.
3. One Y-chromosome located gene PRKY was removed.

**Identification of differentially mutated genes between MET500 and TCGA samples.** Given a gene, we computed the right-tailed p-value to test whether it has significantly higher mutation frequency in metastatic breast cancer samples as follows:

$$p = 1 - \sum_{i=0}^{n} \Pr(i; N, \hat{q}), \qquad (1)$$

where Pr is the probability mass function of binomial distribution, $N$ is the number of genotyped MET500 breast cancer cohorts, $n$ is the number of MET500 breast cancer cohorts in which the gene is mutated and $\hat{q}$ is the mutation frequency of the gene in TCGA dataset (for genes with zero mutation frequency, we used the minimum mutation frequency across all genes). Similarly, we computed left-tailed p-value $(1-p)$ to test whether a gene has significantly lower mutation frequency in metastatic breast cancer samples. To control FDR, we applied the Benjamini–Hochberg procedure on left-tailed and right-tailed p-values, respectively[39]. We noticed that somatic mutations of MET500 cohorts were called by Varscan2 (ref. [40]) while TCGA somatic mutation data hosted on cBioPortal were called by MuTect[30]. To exclude the possibility that the differences inferred between MET500 and TCGA primary tumors were due to pipeline batch effects, we downloaded Varscan2 processed TCGA somatic mutation data from the GDC portal (https://portal.gdc.cancer.gov/) and found gene mutation frequency was highly consistent between the two databases (Supplementary Fig. 11c). In addition, the p-values computed in gene differential mutation analysis (with the formula mentioned above) were also highly correlated (Supplementary Fig. 11d).

**TC analysis with RNA-Seq and microarray data.** To perform TC analysis with RNA-Seq data, we first rank-transformed gene RPKM values for each CCLE cell line and then ranked all the genes according to their rank variation across all CCLE cell lines. The 1000 most-varied genes were kept as "marker genes" (we tried different gene sizes in the early preliminary analysis and did not find the large variation of results, so we decided to choose 1000 most-varied genes in this study). Given RNA-Seq profiles of a cell line (or an organoid) and several patient samples, we compute spearman rank correlation (across the 1000 marker genes) between the cell line (organoid) and each sample and the median value of computed spearman rank correlation values was defined as the transcriptome similarity of the cell line (organoid) with the patient samples. For microarray data, a similar procedure was applied and the 1000 most-varied probe sets were used to compute correlation values.

We also extended the above method to compute CNV similarity. Instead of selecting "marker genes", all of the 1630 commonly genotyped genes were used.

**PAM50 subtyping and t-SNE visualization.** The genefu package was used to determine breast cancer subtype[41,42]. To visualize tumor samples with t-SNE, we first computed the pair-wise distance between every two samples as 1 minus the spearman rank correlation across PAM50 genes and then applied the function Rtsne to perform 2D reduction[43].

**PubMed search.** The number of PubMed abstracts or full texts mentioning a CCLE breast cancer cell line was determined using the PubMed Search feature on 10 May 2018 (https://www.ncbi.nlm.nih.gov/pubmed/). For each cell line, we searched with a keyword "[cell line name] metastasis". We repeated this step for the terms "metastatic", "breast cancer", and "metastatic breast". These searches returned highly correlated results, so we used the search terms which returned the most results: "[cell line name] metastasis".

**Identification of differentially expressed genes and differentially activated gene sets.** DESeq2 was used to identify differentially expressed genes (FDR < 0.001 and abs(log2FC) > 1) and DAVID bioinformatics sever was used to perform Gene Ontology enrichment analysis[44,45]. In our DE analysis, only protein coding genes were considered.

In DA analysis, we first used the R package GSVA to compute ssGSEA scores for the 50 MSigDB hallmark gene sets (http://software.broadinstitute.org/gsea/msigdb/)[46–49]. Then, for each gene set the two-sided Wilcoxon rank-sum test was used to assign the p-value in the comparison of ssGSEA scores between MET500 samples and cell lines (or organoids).

**Software tools and statistical methods.** All of the analysis was conducted in R. The ggplot2 and ComplexHeatmap packages were used for data visualization[50,51]. Tumor purity was estimated using ESTIMATE[16]. CNTools was used to map the segmented CNV data to genes[52]. If not specified, the two-sided Wilcoxon rank-sum test was used to compute the p-value in hypothesis testing.

**Reporting summary.** Further information on research design is available in the Nature Research Reporting Summary linked to this article.

## Data availability
The authors declare that all data supporting the findings of this study are available within the article and its supplementary information files or from the corresponding author upon reasonable request. The data are also available at SYNAPSE (https://www.synapse.org/) under accession number syn18403108. The source data underlying Fig. 2–6 and Supplementary Figs. S1–S8, S10, S11 are provided as a Source Data file.

## Code availability
The code is available at github https://github.com/Bin-Chen-Lab/MetaBreaCellLine.

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

## Acknowledgements

We thank Dr. Nijman of Utrecht Medical Center for providing us organoid RNA-Seq data. We thank Li Huang for helping us create Fig. 1a–e. The research is supported by R21 TR001743 and K01 ES028047 and the MSU Global Impact Initiative. The content is solely the responsibility of the authors and does not necessarily represent the official views of sponsors.

## Author contributions

B.C. and K.L. conceived the study. K.L. performed the majority of computational analysis with the help from S.P., P.A.N., B.S.G., W.Z. D.Z., E.R., and B.C. B.C. supervised the study. All authors contributed to writing, reviewing, and editing the manuscript and approved the manuscript.

## Additional information

**Competing interests:** The authors declare no competing interests.

