## [Peer Review File · Nature Communications]

Reviewers' comments:

Reviewer #1 (Remarks to the Author):

Tumor cell lines in 2D monolayer culture continue to be mainstays of lab-based cancer research. While they are inexpensive and convenient to work with and they have been important for preclinical development of many cancer therapies in current use, the high failure rate for predicting treatment success, and substantial differences from cell lines in 3D culture, organoids, and actual tumor tissue are well-documented. The authors take advantage of published transcriptomes in the CCLC and from other sources to evaluate similarity of commonly used cell lines to a new dataset "MET500" (reference 12) with transcriptomes and genomic profiles from metastases across many tumor sites. In order to make use of CCLC genotype data, the analyses were confined to 1630 genes. Comparison of MET500 and TCGA identified metastasis-enriched gene, a small subset of which were not mutated in any cell line and hence not represented, notably ESR1. Conversely, 32 commonly mutated genes in breast cancer cell lines were only rarely mutated in MET500. Copy number gains, but not CN losses, were more similar between MET500 and CCLC breast lines, and the CN gain similarities were greater for the breast lines derived from metastatic sites. Within breast cancer PAM50 subtypes, CCLC cell lines were identified that best resembled MET500. Similar comparisons were made to patient-derived organoids. Gene set analyses identified pathways associated with differences between MET500 and the cell line and organoid data. A noteworthy result was that MDA-MB-231 cells, widely used to model metastasis, did not resemble basal-like metastatic breast cancer either genomically or transcriptionally.

This work is important in describing and clarifying the relationship of commonly used breast and other tumor cell lines to metastatic human cancer, and makes use of novel MET500 and organoid transcriptome data that have only recently been made public. The description of gene mutations, transcriptional features, and ranked similarity of breast cancer cell lines to human metastatic cancer is biologically interesting, and also has significant practical implications for the breast cancer research community.

1. An unavoidable technical issue is that the MET500 set, TCGA, CCLC, and organoid data were developed from very different patient populations and were analyzed with different methods and different computational pipelines. For example, differences inferred between MET500 and primary tumors in TCGA may be affected by the patient groups these consortia drew from. Furthermore, it appears (lines 360 and 363) that mutations were called by the source data portals, rather than by analysis of primary data by the authors. Are the authors certain that variants seemingly enriched or unique to MET500 would have been called by the CCLC pipeline? Similarly, transcriptional comparisons of MET500, patient-derived organoids, and CCLC are cross-platform (RNA-seq and microarray analysis) or cross-pipeline RNA-seq comparisons (MET500 vs organoids). What steps have

the investigators taken to ensure the validity of these comparisons, as even within platforms batch variation can be an important problem? Is TC analysis the best approach for these comparisons?

2.Line 184. How do these transcriptome matches compare to TCGA (primary tumor) RNA-seq? That is, is there value added by ranking cell lines for similarity to a the metastatic tumor set rather than primary tumors in TCGA, or are the rankings similar?

3.Table 1. How many observations for each mutation?

4.Figure 2a. Which cell line is each? Best to name cell lines or number them so they can be identified by cross-reference to a table.

5.Table S2. Need more information (and citations for source of information) in the figure legend. What does "is.metastatic mean"? Other columns?

6.Line 83. "we identified cell lines that are suitable for modeling metastatic breast cancer samples..." I am not enthusiastic about the use of the word "suitable" here and elsewhere in the manuscript, implying that the identified metastasis-resembling cell lines will perform better as experimental models. This has not been demonstrated. The particular aspects of bulk transcriptome or genotype that are most relevant will likely depend on the experimental question being addressed. With the emphasis on choice of cell lines for modeling, it would be useful to know which of these cell lines forms metastases in mouse xenografts and whether that is connected with similarity to the MET500 set.

7.MDA-MB-231 has been associated with the claudin-low subtype that is not included in PAM50. Would this explain the departure from basal-like?

Reviewer #2 (Remarks to the Author):

This is an interesting manuscript that has applied large scale genomic data from breast cancer cell lines and patient tissue samples to reveal the potential suitability of cell lines to model metastatic

breast cancer. Many interesting findings were described including that cell lines do not harbor mutation patterns similar to metastatic breast cancer samples and speculate that some differences are likely attributable to in vitro culture effects. Of significance is the finding that PAM50 subtypes are maintained in metastatic breast cancer cells. Not surprising is the finding that organoids more closely resemble the transcriptome of patient samples compared to cell lines. The most revealing finding is that the most often used cell line in metastatic breast cancer research is MDA-MB-231, however this cell line is not suitable model for metastasis Also MCF-7 and T47D are not the best choices to represent the Luminal A and B subtypes. Overall while this manuscript may be inaccessible to many researchers due to the data mining complexity, the authors do a good job of distilling the information. Whether the statistical analyses are valid, is beyond my expertise and should be evaluated by a bioinformaticist.

Reviewer #1

1. An unavoidable technical issue is that the MET500 set, TCGA, CCLE, and organoid data were developed from very different patient populations and were analyzed with different methods and different computational pipelines. Differences inferred between MET500 and primary tumors in TCGA may be affected by the patient groups these consortia drew from.

----We thank the reviewer for pointing out this outstanding challenge which commonly exists in meta-analysis work. To mitigate this issue, we employed processed data from well-known studies and also reprocessed raw data using the same pipeline when it was available. In the new revision, we also examined the consistence when data were processed using different pipelines and ensured their consistence.

- a. In the previous paper, Robinson *et al* (authors of the MET500 dataset) performed a pan-cancer comparison of mutation burdens between TCGA and MET500 cohorts¹. In their *Nature* paper, no population bias was reported between the two datasets. The analysis suggests that it is reasonable to compare the somatic mutation data between MET500 and TCGA similarly in this study.

- b. The somatic mutation of MET500 cohorts were called by VarScan2 while our TCGA somatic mutation data was downloaded from cBioPortal, where mutation data was called by MuTect. To exclude the possibility that the differences inferred between MET500 and TCGA primary tumors was due to pipeline batch effects, we downloaded VarScan2 processed TCGA somatic mutation data from the GDC portal (<https://portal.gdc.cancer.gov/>) and found gene mutation frequency was highly consistent between the two databases (cBioPortal and GDC, Fig R1, panel a). In addition, the p-values computed in gene differential-mutation analysis were also highly correlated (Fig R1, panel b). We have added this comparison in Figure S11 (line 450-458).

2. Furthermore, it appears (lines 360 and 363) that mutations were called by the source data portals, rather than by analysis of primary data by the authors. Are the authors certain that variants seemingly enriched or unique to MET500 would have been called by the CCLE pipeline?

----This is a great point.

- a. In gene differential-mutation analysis, we only considered genes that are covered by both whole-exome-sequencing and CCLE Hybrid-capture-sequencing. Therefore, we are certain that

variants seemingly enriched or unique to MET500 have been called by the CCLE pipeline. For example, the gene showing most significant differential mutation frequency between MET500 and TCGA breast cancer samples is ESR1. According to CCLE somatic mutation calling results, it is not mutated in breast cancer cell line while its mutation could be found in other non-breast cancer cell lines (such as HCC1395, a lung cancer cell line).

3. Similarly, transcriptional comparisons of MET500, patient-derived organoids, and CCLE are cross-platform (RNA-seq and microarray analysis) or cross-pipeline RNA-seq comparisons (MET500 vs organoids). What steps have the investigators taken to ensure the validity of these comparisons, as even within platforms batch variation can be an important problem?

- a. The CCLE project profiled the gene expression with *both* RNASeq and microarray. In our analysis, all of the comparisons were performed using the datasets profiled from the same platform (RNASeq vs RNASeq, microarray vs microarray). For RNASeq data, the patient samples were from MET500; for microarray data, patient samples were from GEO. We thank the reviewer for bringing up this important point and have further clarified it in the manuscript (line 164-169).
- b. All of the cell line RNASeq data used in our manuscript was processed by CCLE. The raw data of 55 CCLE breast cancer cell lines are available at GDC Legacy Archive (only two cell lines HMEL and HS274T were missing due to unknown reasons). Therefore, we re-processed the raw RNASeq data of the 55 CCLE breast cancer cell lines using the same pipeline which was used to process MET500 samples. For each cell line, we computed spearman rank correlation between gene expression values quantified by the

two pipelines. Fig R2a shows the distribution of the derived spearman rank correlation values. The median of the distribution is 0.9, suggesting that the gene expression values quantified by the two pipelines are highly consistent. To further demonstrate their consistence, we used MCF7 cell line as an example (Fig R2, panel b). We have added this analysis in Figure S11 (line 407-414).

- c. We re-performed Transcriptome Correlation (TC) analysis between CCLE breast cancer cell lines (with re-processed RNASeq data) and all MET500 breast cancer samples and found that cell line rankings were highly correlated with the results shown in our manuscript (Fig R2, panel c).

- d. It would be ideal to re-process the organoids data with our pipeline and then re-performed the transcriptome comparison between MET500 samples and organoids. Unfortunately, it is not currently feasible for us to accomplish this process in a reasonable timeframe. Specifically, we contacted the authors to ask for the raw data of organoids and their recent response suggested the difficulty to access their data. They responded: **“This will take time and especially with US parties (quite some privacy unfriendly laws have been passed in the last years). In short, yes we can and want to, but we have to abide to laws and regulations. This will likely take quite some time ”** (see Fig R3 for the screen shot of our email communications). However, they highlighted that they did not find any problems when performing bioinformatics analysis with combined organoids and TCGA RNASeq data (without data re-processing). While we plan to perform such an analysis in the future when we can get appropriate access to the data, we believe that this

re-processing of the organoids data would not change the conclusions, based on the results shown in points (b) and (c).

4. Is TC analysis the best approach for these comparisons?

- a. To the best of our knowledge, there is no golden standard computational analysis, although a few studies used the top varying genes for their comparison^{2,3} and we recognize the limitations of this method. According to our previous research, gene-expression based drug discovery did benefit from using cell lines showing high transcriptome similarity with patient samples^{2,4,5} ; therefore, we decided to perform TC analysis and ranked cell lines according to TC analysis results. We have revised the “Discussion” section to discuss more about this (line 367-377).

5. Line 184. How do these transcriptome matches compare to TCGA (primary tumor) RNA-seq? That is, is there value added by ranking cell lines for similarity to the metastatic tumor set rather than primary tumors in TCGA, or are the rankings similar?

- a. We performed TC analysis between CCLE breast cancer cell lines and all TCGA primary tumors. The cell line rankings derived from the two comparisons are correlated with each other (Fig R4).
- b. There are multiple possible reasons why there is a high correlation between the two ranked lists and we do not observe cell lines that have superior transcriptome similarity with metastatic breast cancer samples. **First**, in our TC analysis, the 1000 most-varied genes across CCLE cell lines were used to compute transcriptome similarity between CCLE cell lines and MET500 samples. These genes are informative markers of tissue of

origin while they may not reflect the detailed difference between primary and metastatic tumor samples. However, as mentioned in question 4, there is no golden standard computational analysis. In our research we pay more attention to the utility of cell lines in drug discovery and it has been shown that gene expression profiles are informative biomarkers of drug-response⁶. Therefore, given a drug whose mechanism is not clearly known, it is reasonable to select the cell line which has highest similarity with patient samples to test its effect. However, for researchers who are highly interested in the mechanism of cancer metastasis, there is no requirement for the model to resemble the whole transcriptome as long as it could mimic the key processes in cancer metastasis (such as EMT). Take MDA-MB-231 for example, although its transcriptome similarity with MET500 breast cancer samples is low, it could still be used as a model since it is highly invasive. **Second**, tumor microenvironment has a large impact in shaping the transcriptome of primary and metastatic cancer cells; however, this information is missing when simply culturing cell lines. This may also be the reason why transcriptome of cell lines could not comprehensively reflect the difference between primary and metastatic breast cancers.

- c. In our opinion, the added values of matching transcriptome of cell lines to metastatic tumor samples come from the following two aspects. First, it has been shown that during progression cancer cells experience gradual loss of differentiated phenotype (or cell identity) and undifferentiated primary tumors are more likely to spread to other tissues^{7,8}. Our analysis indicates that metastatic breast cancer cells still retain transcriptome signature indicative of their tissue of origin and this could guarantee the feasibility of

using breast cancer cell lines to model metastatic breast cancer cells. Second, although further investigation is needed, in biopsy-site-specific TC analysis, we showed that cell line ranking derived from the comparison with bone-metastasis samples was different from other biopsy sites (Fig S4). Without using metastatic tumor sample data, neither of the above two observations would be found.

6. Table 1. How many observations for each mutation?

a. We have updated Table S1 to include number of observations for each mutation.

7. Figure 2a. Which cell line is each? Best to name cell lines or number them so they can be identified by cross-reference to a table..

a. Now all the cell line names have been listed in Table S1 according to their order in Figure 2a (from left to right).

8. Table S2. Need more information (and citations for source of information) in the figure legend.

What does “is.metastatic mean”? Other columns?

a. We have revised Table S2 and added annotations for table columns. The column “is.metastatic” indicates whether the cell line is derived from a metastatic site. Since the column name “is.metastatic” is a little bit confusing, we have replaced it with “derived.from.metastatic.site”. In addition, we also added citations for source information (line 633).

9. Line 83. “we identified cell lines that are suitable for modeling metastatic breast cancer samples...” I am not enthusiastic about the use of the word “suitable” here and elsewhere in the manuscript, implying that the identified metastasis-resembling cell lines will perform better as experimental models. This has not been demonstrated. The particular aspects of bulk transcriptome or genotype that are most relevant will likely depend on the experimental question being addressed. With the emphasis on choice of cell lines for modeling, it would be useful to know which of these cell lines forms metastases in mouse xenografts and whether that is connected with similarity to the MET500 set.

- a. We agree with the reviewer on the confusion of using the word “suitable”. We have replaced the sentence with “we identified cell lines that closely resembled the transcriptome of metastatic breast cancer samples of individual subtypes” to reflect our main objective (line 82). In addition, we also revised other sections to avoid the misuse of the word “suitable”.
- b. To the best of our knowledge, currently there is no golden standard to computationally measure how invasive a cell line is. Therefore, for each CCLE breast cancer cell line we computed the ssGSEA score of MSigDB geneset “HALLMARK_EPITHELIAL_MESENCHYMAL_TRANSITION” as a surrogate measure of “invasiveness”. We did not observe positive correlation between the ssGSEA score and transcriptome similarity with MET500 set (Fig R5, panel a). Perhaps it is not surprising that non-Basel-like CCLE breast cancer cell lines (labeled with red color) have higher transcriptome similarity values since most of the MET500 breast cancer samples are non-Basel-like; in addition, Basal-like CCLE breast cancer cell lines have higher ssGSEA scores, which is consistent with known results that they are more invasive.

- c. We repeated the analysis in point (a) to check whether the ssGSEA score of the EMT signature is correlated with similarity to **MET500 Basal-like** samples and did not find the correlation (Fig R5, panel b).

10. MDA-MB-231 has been associated with the claudin-low subtype that is not included in PAM50. Would this explain the departure from basal-like?

- a. We thank the reviewer for helping with the interpretation. Our following analysis revealed this might explain the departure from Basal-like. In a recent *Nature Communications* paper, Nguyen *et al* performed single cell RNASeq on human breast epithelial cells and showed that breast cancer subtypes were associated with cell subpopulations⁹. Gene KRT14 is a hallmark of basal cells, therefore, we checked its expression in MDA-MB-231 cell line from CCLE. To exclude the possibility that the profile of MDA-MB-231 cell line from CCLE might be corrupted, we also collected the RNASeq data of another seven MDA-MB-231 samples from SRA database. We found KRT14 expression in MDA-MB-231 cell line was significantly lower than in MET500 Basal-like breast cancer samples ($p < 0.001$); however, we did not detect significant differential expression of KRT14 between MDA-MB-231 and other subtypes (Fig R6). These results (which have been included the manuscript, line 228-242) suggest that the cell of origin of MDA-MB-231 may not be basal cell and could partially explain the low transcriptome similarity between MDA-MB-231 and MET500 Basal-like samples. As more and more single cell RNASeq data becomes available, it is possible to further explore the cell of origin of claudin-low subtype.

Reviewer #2

This is an interesting manuscript that has applied large scale genomic data from breast cancer cell lines and patient tissue samples to reveal the potential suitability of cell lines to model metastatic breast cancer. Many interesting findings were described including that cell lines do not harbor mutation patterns similar to metastatic breast cancer samples and speculate that some PAM50 subtypes are maintained in metastatic breast cancer cells. Not surprising is the finding that organoids more closely resemble the transcriptome of patient samples compared to cell lines. The most revealing finding is that the most often used cell line in metastatic breast cancer research is MDA-MB-231, however this cell line is not suitable model for metastasis Also MCF-7 and T47D are not the best choices to represent the Luminal A and B subtypes. Overall while this manuscript may be inaccessible to many researchers due to the data mining complexity, the authors do a good job of distilling the information. Whether the statistical analyses are valid, is beyond my expertise and should be evaluated by a bioinformaticist.

- a. Thank you for your positive comments.

Figures

Fig R1. (a) Gene mutation frequency values are highly correlated between cBioPortal and GDC. Each dot is a gene, x-axis represents its mutation frequency (across TCGA Breast Invasive Ductal Carcinoma cohorts) in cBioPortal database and y-axis represents its mutation frequency in GDC database. The solid line represents $y=x$. **(b)** P-values of gene differential-mutation analysis were highly correlated between cBioPortal and GDC. Each dot is a gene, x-axis represents its differential mutation P-value (in $-\log_{10}$ scale) derived from MET500-vs-cBioportal comparison and y-axis represents that derived from MET500-vs-GDC comparison. The solid line represents $y=x$.

Fig R2. (a) The density plot of spearman rank correlation values computed for 55 CCLE breast cancer cell lines. The vertical dashed line indicates median value of the distribution 0.9. **(b)** Gene expression values quantified by two different pipelines are highly correlated in MCF7. Each dot is a gene, x-axis represents its expression value quantified by CCLE pipeline and y-axis represents its expression value quantified by RSEM pipeline. **(c)** Each dot is a CCLE breast cancer cell line, x-axis represents rank value computed with CCLE-processed RNASeq data and y axis represents rank value computed with RSEM-processed RNASeq data. The solid line represents $y=x$.

Fig R3. Screen shot of email communications between us and the author of breast cancer organoids paper.

Fig R4. Each dot is a CCLE breast cancer cell line, x-axis represents its transcriptome similarity with TCGA breast cancer samples and y-axis represents its transcriptome similarity with MET500 breast cancer samples. The solid line represents $y=x$.

Fig R5. (a) Each dot is a CCLE breast cancer cell line, x-axis represents the ssGSEA score of geneset “HALLMARK_EPITHELIAL_MESENCHYMAL_TRANSITION” and y-axis represents transcriptome similarity with all MET500 breast cancer samples. **(b)** Each dot is a CCLE cell line, x-axis represents the ssGSEA score of geneset “HALLMARK_EPITHELIAL_MESENCHYMAL_TRANSITION” and y-axis represents transcriptome similarity with Basal-like MET500 breast cancer samples.

Fig R6. Boxplot of KRT14 expression in MET500 breast cancer samples and MDA-MB-231 cell lines.

References

1. Robinson, D. R. *et al.* Integrative clinical genomics of metastatic cancer. *Nature* **548**, 297–303 (2017).
2. Chen, B., Sirota, M., Fan-Minogue, H., Hadley, D. & Butte, A. J. Relating hepatocellular carcinoma tumor samples and cell lines using gene expression data in translational research. *BMC MEDICAL GENOMICS* **8**, (2015).
3. Vincent, K. M., Findlay, S. D. & Postovit, L. M. Assessing breast cancer cell lines as tumour models by comparison of mRNA expression profiles. *Breast Cancer Research* **17**, 114 (2015).
4. Chen, B. *et al.* Reversal of cancer gene expression correlates with drug efficacy and reveals therapeutic targets. *NATURE COMMUNICATIONS* **8**, (2017).
5. Chen, B. *et al.* Computational Discovery of Niclosamide Ethanolamine , a Hepatocellular Carcinoma Cells In Vitro and in Mice by Inhibiting Cell Division Cycle 37 Signaling. 2022–2036 (2017). doi:10.1053/j.gastro.2017.02.039
6. Wang, Y. *et al.* Systematic identification of non-coding

pharmacogenomic landscape in cancer. *Nature Communications*

doi:10.1038/s41467-018-05495-9

7. Gentles, A. J. *et al.* Machine Learning Identifies Stemness Features Associated with Oncogenic Dedifferentiation Article Machine Learning Identifies Stemness Features Associated with Oncogenic Dedifferentiation. *CELL* 338–354 (2018).
doi:10.1016/j.cell.2018.03.034
8. Chen, H., Lin, F., Xing, K. & He, X. The reverse evolution from multicellularity to unicellularity during carcinogenesis. *Nature communications* **6**, 6367 (2015).
9. Nguyen, Q. H. *et al.* Profiling human breast epithelial cells using single cell RNA sequencing identifies cell diversity. *Nature Communications* 1–12 (2018). doi:10.1038/s41467-018-04334-1

Reviewers' comments:

Reviewer #1 (Remarks to the Author):

As discussed in my earlier review, this work is important in describing and clarifying the relationship of commonly used breast and other tumor cell lines to metastatic human cancer, and makes use of novel MET500 and organoid transcriptome data that have only recently been made public. The description of gene mutations, transcriptional features, and ranked similarity of breast cancer cell lines to human metastatic cancer is biologically interesting, and also has significant practical implications for the breast cancer research community.

The authors have responded substantively to the concerns I identified in the original review, and have made significant changes to the text and analyses provided.

Reviewer #2 (Remarks to the Author):

The comments and revisions provided to reviewer 1 were comprehensive and responsive and addressed key bioinformatic points. As originally state, I am unable to evaluate the biostatistical aspects of the manuscript and now feel comfortable that a rigorous evaluation was conducted. It appears that the revised manuscript is much improved.

Reviewer #3 (Remarks to the Author):

The manuscript compared the mutation, copy number and transcriptome profiles between cancer cell lines, MET500 and TCGA solid tumors and found some interesting findings although most of them are known and expected considering their very different sources, grow environments, and highly heterogeneous nature of tumors. The analyses are fairly comprehensive. However, the practical guidance values in cell line selection from the conclusions may not be that simple. A cell line (such as MDA-MB-231) without similar molecular profile with the selected dataset of metastatic tumors does not void its aggressiveness and high metastatic potential (already so in this case) as each tumor is unique and there are no established molecular features that can reliably predict tumor metastasis. MDA-MB-231 may be in fact not a basal-like subtype (there are some reports) is not collected in MET500 but is still highly aggressive. Metastasis is a complex process and tumor is only part of the story in which host factors may play even a bigger role. I would suggest the authors are a bit more prudent in drawing their conclusions and discuss the limitations of the work.

Additionally, following questions need to be addressed for clarity:

- 1) How "somatic" mutations are called in MET500 and CCLE is not clear and the authors should provide a brief description from cited work. They are important as how to define "somatic" can make a big difference and I guess you can not get true somatic mutations from CCLE without their matching genomic DNA.
- 2) In differential mutation detection between MET500 and TCGA, would stage and cancer subtype affect results? MET500 clearly has higher stages.
- 3) Please provide the 25 highly mutated genes in cell lines (or name a few top ones). They can be artifacts or acquired mutations in cell line culture state (this is related to the question how somatic mutations were called in the cell lines).
- 4) Line 129, is there statistics to measure the more closeness between cell lines of metastatic (vs. non-metastatic) ones with MET500?
- 5) In differential gene expression analysis across three different datasets, would batch effects confound results as clearly they are from different research centers, different library or sequencing protocols? Direct comparisons appear not appropriate without considering these factors.

Reviewer #1

1. As discussed in my earlier review, this work is important in describing and clarifying the relationship of commonly used breast and other tumor cell lines to metastatic human cancer, and makes use of novel MET500 and organoid transcriptome data that have only recently been made public. The description of gene mutations, transcriptional features, and ranked similarity of breast cancer cell lines to human metastatic cancer is biologically interesting, and also has

significant practical implications for the breast cancer research community. The authors have responded substantively to the concerns I identified in the original review, and have made significant changes to the text and analyses provided.

- a. Thank you for your positive comments.

Reviewer #2

The comments and revisions provided to reviewer 1 were comprehensive and responsive and addressed key bioinformatic points. As originally state, I am unable to evaluate the biostatistical aspects of the manuscript and now feel comfortable that a rigorous evaluation was conducted. It appears that the revised manuscript is much improved.

- a. Thank you for your positive comments.

Reviewer #3

1. How “somatic” mutations are called in MET500 and CCLE is not clear and the authors should provide a brief description from cited work. They are important as how to define “somatic” can make a big difference and I guess you cannot get true somatic mutations from CCLE without their matching genomic DNA.

---This is a great point.

- a. The MET500 *Nature* paper¹ stated that “ **Exome libraries of matched pairs of tumor/normal DNAs were prepared for MET500 cohorts and VarScan2 was used to perform mutation analysis** ”, while the CCLE *Nature* paper² stated that “ **Nucleotide substitutions were detected with MuTect using a mode that does not require matching normal DNA and thus identifies all variants**”

that differ from a reference genome”. In our previous revision, we demonstrated the consistency between MuTect and VarScan2.

- b. Without matched normal DNA, germline mutations may be miscalled as somatic mutations and this could be a potential issue in the following study. Therefore, in our analysis we used the filtered version of CCLE mutation MAF file (CCLE_hybrid_capture1650_hg19_NoCommonSNPs_NoNeutralVariants_CDS_2012.05.07.maf) in which common polymorphism variants have been excluded.
- c. We have revised the method section and made a more clear description about how the somatic mutations were called in the two datasets (line 415).

2. In differential mutation detection between MET500 and TCGA, would stage and cancer subtype affect results? MET500 clearly has higher stages.

----We thank the reviewer for raising this interesting biological question.

- a. We agree that differential mutation between MET500 and TCGA may depend on various clinical/molecular features such as age, gender, stage and cancer subtype. Considering all these factors requires many more samples as well as a multitude of computational analyses, which we think it is beyond the scope of this work.
- b. We agree that MET500 samples have higher stages. To exclude the possibility that the identified differential mutation between MET500 and TCGA are confounded by stages, we further performed differential mutation analysis between MET500 samples and TCGA samples at different late stages. There are three stages with more than 100 sequenced TCGA samples: Stage IIA (261), Stage IIB (169), Stage IIA (105). Notably, we found that the p-values derived from the above three

comparisons were highly correlated with that derived from MET500 vs TCGA (full set) comparison, suggesting that the stage information is not confounding the differential mutation analysis (Fig R1, panel a, b and c).

3. Please provide the 25 highly mutated genes in cell lines (or name a few top ones). They can be artifacts or acquired mutations in cell line culture state (this is related to the question how somatic mutations were called in the cell lines).
 - a. We have updated Supplementary Data 1 to include the highly mutated genes in cell lines.

4. Line 129, is there statistics to measure the more closeness between cell lines of metastatic (vs. non-metastatic) ones with MET500?
 - a. For each of the 109 genes with high copy-number-gain in MET500 dataset, we computed two values to measure its CNV difference between MET500 and cell lines:
 $D1 = \text{abs}(\text{median.across.MET500} - \text{median.across.cell.line.of.non.metastatic}),$
 $D2 = \text{abs}(\text{median.across.MET500} - \text{median.across.cell.line.of.metastatic}).$
Fig S1e indeed shows the distribution of the two statistics and Wilcoxon rank test indicates that D2 is significantly smaller than D1. In the new revision, we have revised the text to clarify this point (line 754, legend of Supplementary Figure 1e).

5. In differential gene expression analysis across three different datasets, would batch effects confound results as clearly they are from different research centers, different library or sequencing protocols? Direct comparisons appear not appropriate without considering these factors.

---We appreciate that the reviewer raised this concern. We highlighted this issue in the second revision, where we discussed the computational approach to minimize batch effect. In our recent paper³, we justified that two different datasets (TCGA and GTEx) could be integrated to identify differentially expressed genes. Nevertheless, we performed the analysis using RUVg⁴, a popular package to normalize RNA-Seq data from different studies, and found our conclusion still holds.

- a. We re-performed each of the four DE comparisons shown in Fig S9 and the only difference with our previous comparison is that RUVg-inferred confounding factor values were plugged into DESeq2 for model fitting (the number of confounding factor is 1, all genes whose adjusted p-value LARGER than 0.01 in our previous DE analysis were considered as control genes). We finally identified 749 subtype-and-model independent DE genes (738 up-regulated, 11 down-regulated), all of which were among our previous-identified 1017 subtype-and-model independent DE genes. The results suggest that our previous DE analysis were not dominated by batch factors.
- b. It may not be surprising that we found fewer DE genes after considering RUVg-inferred factors since the biological effects here are confounded with batch effects. When removing the variation caused by batch factors, we also took the risk of removing the variation of interest. Moreover, as we changed the number of control genes in RUVg, the number of DE genes would be changed. Therefore, we would keep the results from the original analysis.
- c. We revised the manuscript to discuss the results of the RUVg analysis as well.(Line 290)

Figures

Figure R1. (a) (b) (c) Scatter plot of gene differential-mutation analysis p-values (in $-\log_{10}$ scale). Each dot is a gene, x-axis represents p-value derived from MET500 vs stage-specific TCGA cohorts comparison and y-axis represent p-values derived from MET500 vs full TCGA cohorts comparison.

References

1. Robinson, D. R. *et al.* Integrative clinical genomics of metastatic cancer. *Nature* **548**, 297–303 (2017).
2. Barretina, J. *et al.* The Cancer Cell Line Encyclopedia enables predictive modelling of anticancer drug sensitivity (vol 483, pg 603, 2012). *NATURE* **492**, 290 (2012).
3. Zeng, W. Z. D., Glicksberg, B. S., Li, Y. & Chen, B. Selecting precise reference normal tissue samples for cancer research using a deep learning approach. *BMC medical genomics* **12**, 21 (2019).
4. Risso, D., Ngai, J., Speed, T. P. & Dudoit, S. Normalization of RNA-seq data using factor analysis of control genes or samples. *Nature biotechnology* **32**, 896 (2014).

Figure R1. (a) (b) (c) Scatter plot of gene differential-mutation analysis p-values (in $-\log_{10}$ scale). Each dot is a gene, x-axis represents p-value derived from MET500 vs stage-specific TCGA cohorts comparison and y-axis represent p-values derived from MET500 vs full TCGA cohorts comparison.

REVIEWERS' COMMENTS:

Reviewer #3 (Remarks to the Author):

Thank the authors for addressing the concerns and revising the manuscript, which made it more clear not just for me but also for potential readers.